

# Improvements to stratospheric chemistry scheme in the UM-UKCA (v10.7) model: solar cycle and heterogeneous reactions

Fraser Dennison[1], James Keeble[2,3], Olaf Morgenstern[1], Guang Zeng[1], N. Luke Abraham[2,3], and Xin Yang[4]

[1]National Institute of Water and Atmospheric Research, Wellington, New Zealand
[2]Centre for Atmospheric Science, Department of Chemistry, University of Cambridge, Cambridge, UK
[3]National Centre for Atmospheric Science, UK
[4]British Antarctic Survey, Cambridge, UK

**Correspondence:** Fraser Dennison (fraser.dennison@niwa.co.nz)

**Abstract.**

Improvements are made to two areas of the United Kingdom Chemistry and Aerosol (UKCA) module, which forms part of the Met Office Unified Model (UM) used for weather and climate applications. Firstly, a solar cycle is added to the photolysis scheme. The effect on total column ozone of this addition was found to be around 1–2% in mid-latitude and equatorial regions in phase with the solar cycle. Secondly, reactions occurring on the surfaces of polar stratospheric clouds and sulfate aerosol are updated and extended by modification of the uptake coefficients of five existing reactions and the addition of a further eight reactions involving bromine species. These modifications are shown to reduce the overabundance of modeled total-column ozone in the Arctic during October to February, southern mid-latitudes during August, and the Antarctic during September. Antarctic springtime ozone depletion is shown to be enhanced by 25 DU on average, which now causes the ozone hole to be somewhat too deep compared to observations. We show that this is in part due to a cold bias of the Antarctic polar vortex in the model.

## 1 Introduction

Stratospheric chemistry is a crucial aspect of chemistry–climate models primarily due to the coupling of ozone with atmospheric dynamics. Ozone strongly absorbs ultraviolet (UV) radiation, thus controlling the temperature of the stratosphere and hence the speed and structure of large scale stratospheric circulation (e.g. McLandress et al., 2010; Braesicke et al., 2013; Keeble et al., 2014). Ozone depletion in southern high latitudes has been shown to drive trends in the Southern Annular Mode (SAM) which is the leading mode of climate variability in the Southern Hemisphere (SH) (e.g., McLandress et al., 2011; Thompson et al., 2011; Dennison et al., 2015). In turn the SAM affects numerous other climate features, e.g. Antarctic surface temperatures (Thompson and Solomon, 2002; Marshall, 2007; Gillett et al., 2006), the Southern Ocean storm track (Yin, 2005), the atmospheric blocking frequency (Dennison et al., 2016), and sea ice (Hall and Visbeck, 2002; Sen Gupta and England, 2006). In northern high latitudes, chemical ozone loss is less pronounced (e.g. Solomon et al., 2014), although there





is some evidence of the effect of strong ozone depletion events on the large-scale circulation, with impacts on mid-latitude weather (e.g. Shindell et al., 2001; Ivy et al., 2017).

The abundance of ozone is affected by solar radiation via photochemical reactions. The dominant mode of ozone production is via the photolysis of oxygen; ozone destruction is also initiated by photolysis. Ozone-depleting reactions involving only

oxygen compounds are known as the Chapman cycle (Chapman, 1930). Additionally, there are photolysis reactions which are part of various ozone depleting catalytic cycles, for example the photolysis of chlorine peroxide ($Cl_2O_2$) (Crutzen, 1974; Molina and Rowland, 1974; McGrath et al., 1990). Solar radiation varies on an approximately 11-year cycle (Solanki et al., 2013, and references therein). This cycle is known to have a significant effect on stratospheric ozone (e.g. Zerefos et al., 1997; Calisesi and Matthes, 2006; Tourpali et al., 2007; Kuroda et al., 2008; Gruzdev, 2014). Specifically, studies such as Dameris

et al. (2006), Steinbrecht et al. (2004), and Keeble et al. (2018) make the point that ozone increases in the early 2000s were caused by the Sun going through its solar maximum and did not constitute evidence of ozone recovery due to reductions in stratospheric halogen.

The importance of simulating the solar cycle in chemistry-climate models (CCMs) has been noted by a number of studies (e.g. Egorova et al., 2005; Langematz et al., 2005; Tourpali et al., 2003; Labitzke et al., 2002). Of the 20 CCMs participating in

the first phase of the Chemistry-Climate Model Initiative (CCMI-1), only three did not consider solar variability (Morgenstern et al., 2017) . In this paper we describe the addition of a solar cycle to the photolysis scheme in a newer version of that model and show the impact this has on modeled stratospheric ozone. In the following we will refer to the model as "UM-UKCA" (Unified Model - United Kingdom Chemistry and Aerosols) which forms part of the United Kingdom Earth System Model (UKESM) and other constellations of the Met Office Unified Model (UM).

In addition to photolysis reactions, heterogeneous reactions are also important for stratospheric ozone. Halogen species such as hydrogen chloride (HCl), chlorine nitrate ($ClONO_2$), hydrogen bromide (HBr), and bromine nitrate ($BrONO_2$) react on the surfaces of stratospheric aerosols, e.g. polar stratospheric clouds (PSCs), forming species which are subsequently photolysed into ozone depleting radicals. In this study, two configurations of UKCA are used which differ in the representation of stratospheric chemistry. In previous versions of the UKCA chemistry scheme, only five heterogeneous reactions are considered

(Morgenstern et al., 2009), none of which involve the activation of bromine. While bromine is much less abundant than chlorine in the stratosphere, on a per-atom basis it is more effective at depleting ozone (Daniel et al., 2007; Newman et al., 2007; Sinnhuber et al., 2009). As such, it has been shown to have a significant effect in some instances. For example, Sinnhuber et al. (2009) have estimated that approximately half of northern mid-latitudes ozone loss over the period 1980-2005 may be attributable to anthropogenic bromine emissions. To account for this, in order to improve the simulation of stratospheric ozone

in the UKCA model, eight further heterogeneous reactions involving bromine species are added. In addition, rate constants for the heterogeneous reactions are updated to match latest literature values.

This study details the effects on simulated stratospheric ozone of adding solar variability to the UKCA photolysis scheme, and of expanding heterogeneous chemistry to include reactions involving bromine compounds. These two additions will be examined separately. Section 2 gives a brief description of the UM-UKCA model. Section 3 deals with the photolysis and

Section 4 deals with the heterogeneous chemistry. Each section details the changes made to the model, the experiments run to





test the impact of theses changes, the effect of the changes on the stratospheric ozone and discussion of the results. Section 5 provides a summary.

## 2    Model Description

The United Kingdom Chemistry and Aerosol (UKCA) module is part of the Met Office Unified Model (UM). Here we use a
configuration with a free running atmosphere and prescribed sea ice and sea surface temperatures from the HadISST dataset (Rayner et al., 2003) and operate the model at a resolution of $1.875°$ longitude by $1.25°$ latitude with 85 levels extending to 85 km. The UKCA model contains a number of chemistry configurations; here we use the combined stratosphere and troposphere chemistry (CheST) option (a combination of Morgenstern et al. (2009) and O'Connor et al. (2014)) in a GA7.1 (Walters et al., 2017) atmosphere configuration identical to that used in Esentürk et al. (2018) but at Unified Model version 10.7. This
configuration includes 75 chemical species and 283 reactions. Further details on the relevant model chemistry are provided in Sections 3 and 4.

## 3    Photolysis and the Solar Cycle

### 3.1    Model

Photolysis rates in the configuration of the UKCA model used here are calculated using a combination of the FAST-JX scheme
(Wild et al., 2000; Bian and Prather, 2002; Neu et al., 2007) and look-up tables. FAST-JX covers wavelengths from 177 to 850 nm over 18 wavelength bins, thus making it suitable for the simulation of chemistry in the stratosphere. FAST-JX calculates scattering for all wavelength bands. The implementation of FAST-JX in UKCA is described by Telford et al. (2012). Above about 60 km, wavelengths shorter then 177 nm become important. In this region, UKCA additionally uses a look-up table of photolysis rates (Lary and Pyle, 1991; Morgenstern et al., 2009) for the contribution to photolysis occurring at these
wavelengths. Rates from the two schemes are added together to form the full photolysis rate.

The implementation of the solar cycle in the model uses solar irradiance data from Lean et al. (2005). Using a singular-value decomposition analysis, solar irradiance $I$ (Wm$^{-2}$, which comes as a function of time and wavelength) is expressed as

$$I(\lambda, t) = I_0(\lambda) + I_1(\lambda)T(t) \tag{1}$$

i.e. a constant term $I_0$ (the temporal average over the time series) and the product of a spectrally varying term $I_1$ and a
time series $T$ at monthly resolution. $I_1$ and $T$ are then normalized such that the standard deviation of $T$ equals 1 and $I_1$ is predominantly positive. These terms are illustrated in Figure 1. The decomposition captures the full variability in the Lean et al. (2005) data extremely well. The spectral term $I_1$ is processed by the FAST-JX binning algorithm into an 18 element array (Bian and Prather, 2002) and for the lookup table part of photolysis it is processed into a 46-element array covering the wavelength range of 116.7 to 176.2 nm (the Lean et al. (2005) data cover wavelengths of 120 nm and above, so the first five





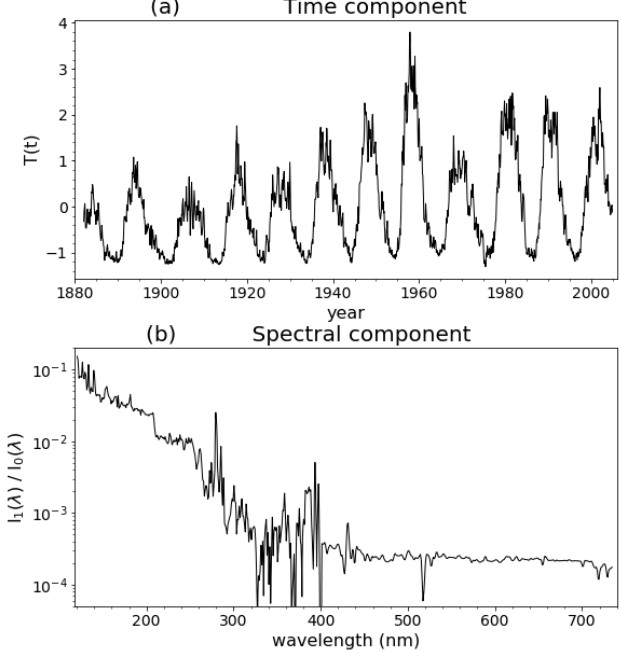

**Figure 1.** Singular value decomposition of the solar irradiance data into the (a) time series and (b) spectral components.

elements are zero). The top-of-the-atmosphere incoming shortwave flux is then modified by these time series and wavelength factors. The time series is extended into the future by repeating an average of the last five solar cycles.

To test this modification, two 15-year runs were produced: one with the solar cycle switched on and a control run with the solar cycle switched off. These runs use historic GHG and ODS forcing. The effect of the solar cycle was tested by comparing the oxygen photolysis rates in the two runs. This reaction is a component of the cycle which is the main mode of ozone production in the stratosphere and requires radiation with wavelengths shorter than 240 nm (Chapman, 1930). Figure 1 shows that at these wavelengths the solar irradiance will vary with a amplitude of up to $10\%$ for a 1-standard deviation anomaly of solar output $I$, so we expect an effect on the oxygen loss rate of a similar magnitude.

## 3.2 Results

Figure 2(a) shows the anomalous globally averaged oxygen loss rate in the solar cycle run relative to the control run. The 11-year cycle is clearly evident with the amplitude of approximately $\pm10\%$ in the mesosphere. This magnitude is what is expected given the spectral component shown in Figure 1 is of the order of $10\%$ for the shorter wavelengths at which this reaction occurs. The amplitude of the anomaly decreases in the stratosphere as this short-wavelength radiation gets attenuated.

The effect of the solar cycle implementation on ozone is shown in Figure 2(b). The percentage difference in total column ozone (TCO), averaged over 60°S–60°N, in the solar cycle run relative to the control run is indicated by the cyan curve; a 3-year running mean is also shown (blue). It can be seen that ozone varies in phase with the solar cycle, illustrated here by the





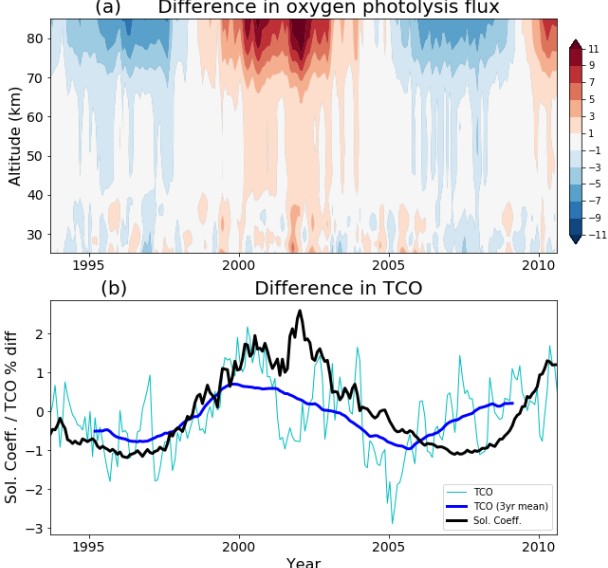

**Figure 2.** (a) The global mean anomaly in the flux of the oxygen photolysis reaction and (b) the mean total column ozone anomaly $60°S$–$60°N$ (cyan, and 3-year running mean in blue), both expressed as percentage difference between the control and solar cycle runs. Overlaid on (b) is the solar coefficient time series, $T$ (black).

black curve (taken from Figure 1(a)), with an amplitude of around 1% or 2–3 Dobson Units (DU). We only plot the results for the $60°S$–$60°N$ region here because the variability in ozone is quite large in the polar regions, masking the impact of solar variability. In the northern polar region, the amplitude of the cycle in ozone difference is around 4%, whereas in the southern polar region, meteorological variability of ozone masks any influence of solar variability on ozone in a simulation of only 15 years in length.

### 3.3 Discussion

The magnitude of the solar cycle driven ozone effect shown here i.e., 1–2% between solar maximum and solar minimum, compares well to other observational and climate model studies. A review of a number of observational studies by Calisesi and Matthes (2006) found a magnitude around 2% for stratospheric ozone. Reinsel et al. (2002), using total column ozone data from the Total Ozone Mapping Spectrometer (TOMS) satellite instruments, found the effect to be 1-2%. More recently, Maycock et al. (2016) found the effect in the lower stratosphere to be around 1% using the Stratospheric Aerosol and Gas Experiment (SAGE) and Solar Backscatter Ultraviolet Merged Ozone Dataset (SBUVMOD) observations. Maycock et al. (2017) examine a number of CCMI models and found them to be largely consistent with these observational findings.

Figure 2 suggests there may be some phase shift between the solar cycle and its effect on stratospheric ozone. Although, the model runs presented here are too short to verify this, such phase shifts have been identified in the observations. For example, Gruzdev (2014) found that in the middle and lower stratosphere the oscillation of ozone leads the solar cycle while, in the

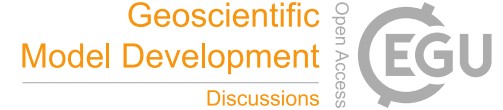

upper stratosphere it lags the solar cycle, with the difference being as much as a quarter cycle. It is proposed that these results follow from the variation in the strength of the Brewer-Dobson circulation with the solar cycle (Kodera, 2006). Angell (1989) also found that ozone leads the solar cycle but with a much smaller phase difference of around half a year.

Bednarz et al. (2018) also examine an implementation of a solar cycle in the UKCA model. They take a slightly different approach wherein the amplitude of the solar cycle is defined by the difference between the 1981 solar maximum and the 1986 solar minimum. Observational data are used to partition this variation across the 18 FAST-JX wavelength bins as well as the UM shortwave radiation scheme's six bins. No spectral variation in the solar irradiance is accounted for in the look-up table. The results presented here are broadly similar to those shown by Bednarz et al. (2018); they report a solar driven effect on total column ozone in the 60°s - 60°N region of 4.1 ($\pm$1.9) and 5.6 ($\pm$1.4) DU/Wm$^{-2}$ using two different methods, where the difference in total irradiance between solar maximum and minimum was 1.06Wm$^{-2}$. The difference between solar maximum and minimum shown in Figure 2 expressed as an absolute difference is around $4-5$ DU. Bednarz et al. (2018) also find the ozone response to increase away from the equator but do not report values for the polar regions.

The approach we have taken here has the advantage of using historical solar data. Figure 1(a) shows that the amplitude of the cycle is not constant over the historical record, and as such historical simulations should be more accurate than if using a cycle of constant amplitude. Also, our more comprehensive treatment of the look-up table component, which is used here for pressure levels above 20 Pa, is likely to produce better results in the upper model levels, for example, in the photolysis of oxygen shown in Figure 2(a). Our solar cycle implementation is solely in the UKCA module, unlike Bednarz et al. (2018) who also implemented solar variability in the UM shortwave radiation scheme (which is separate from the photolysis scheme). It may be possible to extend our treatment of solar variability to the UM radiation scheme although it is unlikely to have a large effect given the smaller amplitude of the solar cycle at the relevant wavelengths. Bednarz et al. (2018) find that the difference between solar maximum and minimum in the shortest UM wavelength band (covering 200-320 nm) is only 0.56%, and is less than 0.1% in the longer-wavelength bands.

## 4 Heterogeneous Chemistry

### 4.1 Model

The pre-existing UKCA stratospheric scheme included five heterogeneous reactions, the first five listed in Table 1. Reaction rates are calculated based on uptake on three classes of aerosols: ice, nitric acid trihydrate (NAT), and sulphate aerosol (SA). Ice is taken from the UM microphysics scheme (Wilson and Ballard, 1999), i.e. it is consistent with cloud physics used elsewhere in the model, the calculation of NAT abundances follows Hanson and Mauersberger (1988), and a climatology of sulphate aerosol based on year 2000 is prescribed (Morgenstern et al., 2010). Table 1 lists the uptake coefficient on each aerosol type for each reaction; the uptake coefficient is the probability of the gas molecule undergoing an irreversible reaction upon collision with the specified surface. Uptake coefficients for these reactions are those recommended by the Jet Propulsion Laboratory (JPL) (Burkholder et al., 2015), in many cases this represents a change from the coefficients in pre-existing scheme (which are listed under "Old" in Table 1).



**Table 1.** Heterogeneous reactions included in the updated UKCA module and the uptake coefficients on ice, NAT and SA.

| | Uptake Coefficient | | | | | |
| | Ice | | NAT | | SA | |
| Reaction | Old | New | Old | New | Old | New |
| --- | --- | --- | --- | --- | --- | --- |
| $ClONO_2 + HCl \rightarrow 2\,Cl + HNO_3$ | 0.3 | $0.3^1$ | 0.3 | $0.2^1$ | | $f^2$ |
| $ClONO_2 + H_2O \rightarrow HOCl + HNO_3$ | 0.3 | $0.3^1$ | 0.006 | $0.004^1$ | $f^5$ | $f^2$ |
| $HOCl + HCl \rightarrow 2\,Cl + H_2O$ | 0.3 | $0.2^1$ | 0.3 | $0.1^1$ | $f^6$ | $f^2$ |
| $N_2O_5 + H_2O \rightarrow 2\,HNO_3$ | 0.03 | $0.02^1$ | 0.0006 | $0.0004^1$ | 0.1 | $0.1^3$ |
| $N_2O_5 + HCl \rightarrow Cl + NO_2 + HNO_3$ | 0.03 | $0.03^1$ | 0.003 | $0.003^1$ | | |
| $HOBr + HCl \rightarrow BrCl + H_2O$ | | $0.25^4$ | | | | |
| $BrONO_2 + HCl \rightarrow BrCl + HNO_3$ | | $0.3^4$ | | | | $0.9^1$ |
| $BrONO_2 + H_2O \rightarrow HOBr + HNO_3$ | | $f^4$ | | | | $f^4$ |
| $HOBr + HBr \rightarrow 2\,Br + H_2O$ | | $f^4$ | | | | |
| $HOCl + HBr \rightarrow BrCl + H_2O$ | | $f^4$ | | | | |
| $ClONO_2 + HBr \rightarrow BrCl + HNO_3$ | | $f^4$ | | $f^4$ | | |
| $BrONO_2 + HBr \rightarrow 2\,Br + HNO_3$ | | $f^4$ | | | | |
| $N_2O_5 + HBr \rightarrow Br + NO_2 + HNO_3$ | | | | $f^4$ | | |

Uptake coefficients are denoted $f$ where they are not universal constants, see the references for the full formulation.

[1] Burkholder et al. (2015)

[2] Shi et al. (2001) (as recommended by both JPL and IUPAC)

[3] This was retained from the original because the JPL recommendation, which is a function of temperature, evaluates to approximately 0.1 for typical stratospheric values

[4] Crowley et al. (2010)

[5] Steele and Hamill (1981); Zhang et al. (1994); Cox et al. (1994)

[6] Zhang et al. (1994)

We extend the heterogeneous chemistry scheme by adding eight reactions involving bromine species, these are also listed in Table 1. The uptake coefficients for the additional reactions follow International Union of Pure and Applied Chemistry (IUPAC) (Crowley et al., 2010) recommendations except for the $BrONO_2 + HCl$ reaction on SA surfaces which follows Burkholder et al. (2015).

5 The changes to the heterogeneous chemistry are initially tested in separate parts to assess their relative importance, namely the updating of the NAT and ice uptake coefficients for the original five reactions (labeled "New Coeff."), the new formulation for the uptake coefficients on sulfate aerosol ("Shi 2001") and the addition of the eight new bromine reactions ("New Br"), as well as these modifications in combination ("All"). 15-month simulations are produced for each and compared to a control simulation with the original heterogeneous chemistry. In order to examine only the direct effect of the modifications, feedbacks
10 between the chemistry (ozone, methane, nitrous oxide and water vapour) and radiation have been switched off for these runs (i.e. these runs are identical in terms of their meteorological variables).





Additionally, a 21-year run (1989–2009) for both the original and new configuration is produced, with chemistry-radiation feedbacks switched on, to provide a more reliable indication of the impact on stratospheric chemistry. These runs are forced by the IPCC "historic" and RCP 6.0 greenhouse gas scenario (Masui et al., 2011; Meinshausen et al., 2011; van Vuuren et al., 2011) and the A1 scenario for ozone depleting substances (WMO, 2011). We compare these runs to the National Institute of Water and Atmospheric Research - Bodeker Scientific (NIWA-BS) total column ozone dataset (version 3.4, see http://www.bodekerscientific.com/data/total-column-ozone). The time span we use for comparison is 1996–2009, excluding 2002. The beginning of the run is not used to ensure any transient effects of the chemistry changes are discounted. The year 2002 is not included because the stratospheric sudden warming that occurred in that year over Antarctica resulted in anomalously low polar ozone in the observations.

## 4.2 Results

The largest effect of the changes to the heterogeneous chemistry is found at southern polar latitudes. Figure 3(a) shows the differences in total column ozone, averaged over the region south of 65°S, between the various sensitivity runs and the control run. Updating the NAT and ice uptake coefficients of the original five reactions (yellow curve) increases ozone during spring. This is as expected given that the coefficients were either unchanged or decreased from the original version, effectively decreasing chlorine activation. The magnitude of this change is up to 9 DU and occurs in late September. This change in the NAT and ice coefficients is outweighed by adopting the Shi et al. (2001) formulation for the uptake coefficients on sulphate aerosol (blue curve). The effect of this change is an enhancement of the springtime ozone depletion, peaking at 20 DU in early October. By contrast, the effect of the additional bromine reactions (green curve) is evident from the beginning of the run and amounts to around 10 DU in late winter, before the pronounced springtime ozone depletion begins. The additional bromine reactions are at maximum effect in early spring where they amount to 20 DU of additional ozone depletion. These three changes are combined in the run labeled "All" (red curve), which shows the maximum effect on total column ozone to be a decrease in ozone of 30 DU in late September/early October, but due to the inclusion of the bromine reactions is important in all seasons. The effect from the combination of the three modifications is slightly more than the sum of the individual runs. This effect is confined mostly to September and is up to about 4 DU. Figure 3(b) shows the equivalent for the Northern Hemisphere. The overall impact on ozone is smaller (at maximum 15 DU during March) and it is notable that the addition of bromine is producing the majority of the effect relative to the either of the modifications to the original reactions.

For the rest of this section we focus on the fully interactive simulations, comparing the new heterogeneous chemistry ("New Het.") to the control simulation and to the NIWA-Bodeker total column ozone dataset. Figure 4 shows the zonal mean total column ozone climatology in each of the model runs. The figure uses crosshatching and stippling to illustrate areas where the model underestimates or overestimates, respectively, the NIWA-BS dataset by more than 30 DU. There are no NIWA-BS data over the winter poles, these areas are marked by the dashed contours. Comparing the two runs, it can be seen that the new heterogeneous chemistry reduces biases in a number of areas: the northern high latitudes during October to February, the equatorial region during August to October, the southern mid-latitudes during August and the southern polar region during the September onset of the ozone hole. The one area where the changes to the heterogeneous chemistry make the model





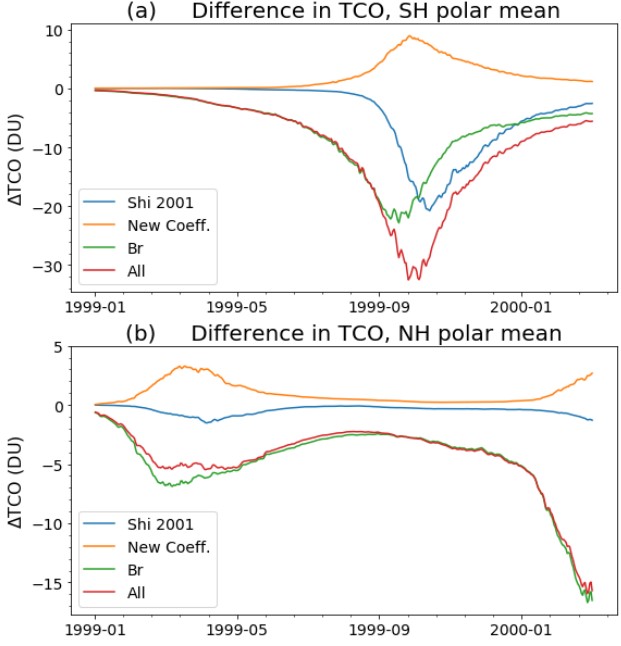

**Figure 3.** The difference in mean total column ozone (a) south of $65°$S and (b) north of $65°$N, between the various sensitivity runs and the control run.

notably worse is the southern polar region during austral summer where the existing underestimation of ozone by the model is exacerbated. This can be seen more clearly in Figure 5 which shows the polar mean (south of $65°$S) total column ozone climatology. During springtime, peak ozone depletion is enhanced by around 25 DU (note: this is slightly smaller than that indicated in Figure 3, likely reflecting that the 1999 forcings during the sensitivity runs were more conducive to ozone depletion

5    than the 1996–2009 average). This increased depletion persists with a similar magnitude through summer. The effect of the new configuration is that the extent of peak ozone depletion is now overestimated relative to the NIWA-BS dataset, whereas for the control run it was underestimated. An issue with the control run illustrated here is the slow summertime replenishing of ozone. Although, "New Het." replenishes ozone at a similar rate, the increase in the depth of the ozone hole means that the summertime discrepancy in ozone increases significantly relative to NIWA-BS.

10    The overestimation of springtime ozone depletion shown by Figure 4 is linked to model temperature biases which influence polar ozone depletion. We illustrate this in Figure 6, which shows the relationship between polar total column ozone and polar lower stratospheric mean temperature (south of $65°$S and pressure levels between 100–10 hPa) for October in the model runs and the NIWA-BS dataset (with temperatures from the ERA-Interim reanalysis (Dee et al., 2011)). The model has a substantial cold bias relative to the ERA-Interim temperature which is not solely due to differences in ozone. Based on the gradient of the

15    ozone-temperature relationship shown here, it is likely that a hypothetical correction of the temperature bias would reveal a superior simulation of the ozone depletion by the "New Het." configuration.





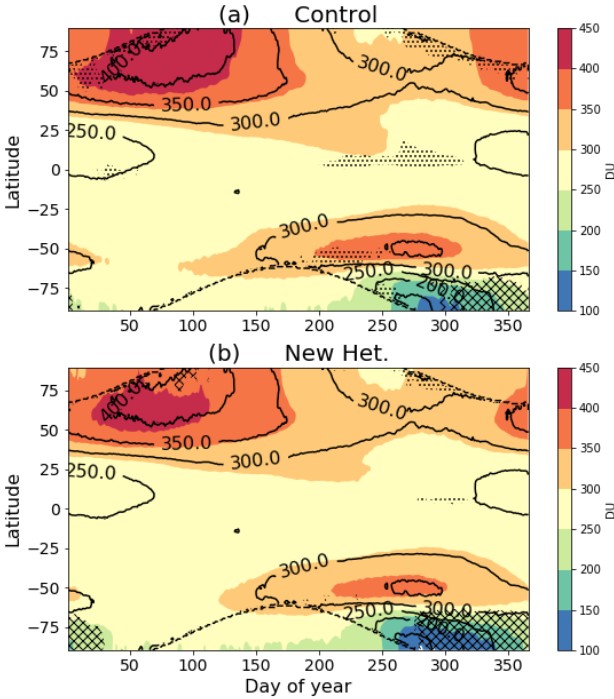

**Figure 4.** Zonal mean total column ozone climatology (1996–2009, excl. 2002) of the (a) control and (b) "New Het." runs. Contours show the NIWA-BS data, crosshatching/stippling indicates where the model under/overestimates NIWA-BS data by more than 30 DU. Dashed contours mark the absence of NIWA-BS data.

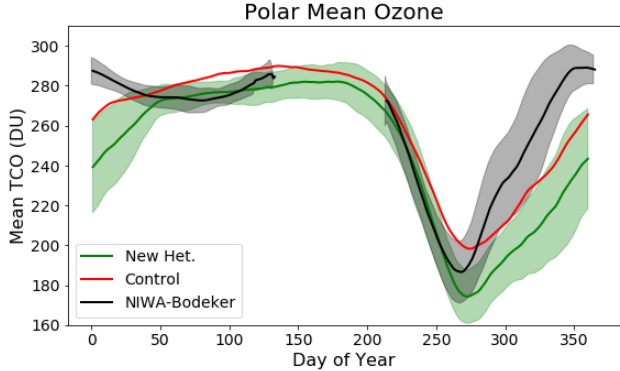

**Figure 5.** Southern polar mean total column ozone climatology of the "New Het." (green) and control (red) runs as well as the NIWA-BS data (black). The shaded bands show ±1 standard deviation, uncertainty in the control run is not shown here for clarity. The average is over the region south of $65°$S and for the years 1996–2009 (excl. 2002).

Figure 7 shows the difference in zonal mean ozone mixing ratio during October. Student's $t$ test is used to identify regions where the model runs differ significantly. The enhanced polar ozone depletion is primarily at the lower reaches of the ozone



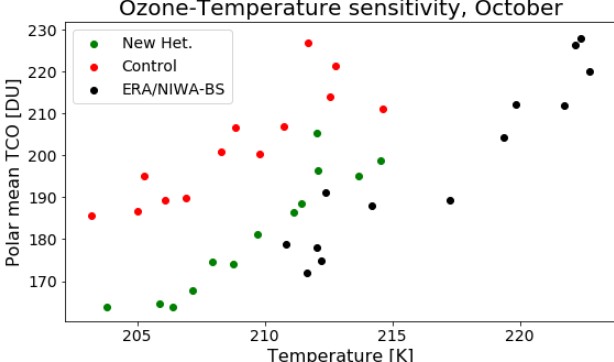

**Figure 6.** Southern polar mean total column ozone versus the mean temperature in the polar lower stratosphere for the "New Het." (green) and control (red) runs as well as NIWA-BS / ERA-Interim (black). The average is taken over the region south of 65°S; temperature is additionally averaged over the 100–10 hPa pressure range. Data cover the years 1996–2009 (excl. 2002).

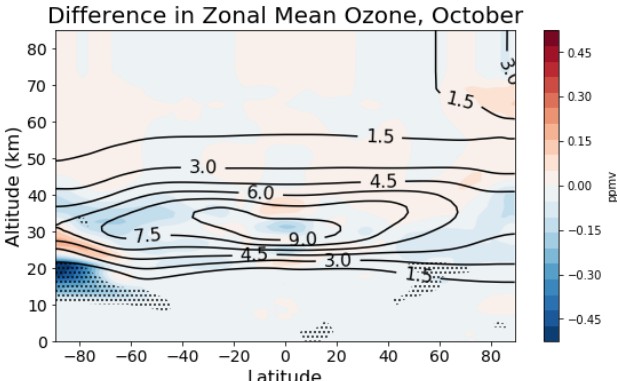

**Figure 7.** Difference in zonal mean ozone between the "New Het." and control runs during October 1996–2009 (shading). Contours illustrate the control run climatology. Stippling indicates the difference is significant according to Student's $t$ test (p<0.01).

layer between around 12 to 24 km in altitude. Figure 8 shows the change in flux of the various ozone production/destruction pathways in this southern polar lower stratosphere region. The largest change is via the $Cl_2O_2$ photolysis cycle. This change represents an 8% increase over the flux in the control run.

Table 2 lists the differences in flux of the various heterogeneous reactions between the control and "New Het." runs. The

5     changes in the original five reactions include an increase in the rate of the $ClONO_2 + HCl$ reaction at the expense of the hydrolysis reaction. This is almost entirely due to the change in the calculation of the uptake coefficient on sulphate aerosol. Also, the rate of $N_2O_5$ hydrolysis decreases; in this case the uptake rates on both ice and NAT have been lowered by a third (while, the reaction on sulfate is unchanged). The most important of the new bromine reactions are $BrONO_2 + HCl$ and $BrONO_2 + H_2O$, the two bromine reactions for which reaction on sulphate aerosols is considered.





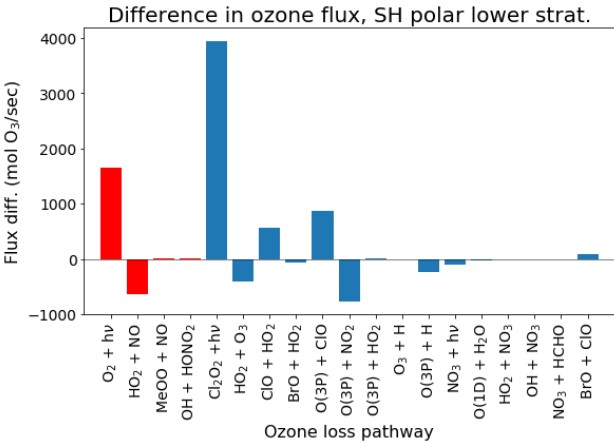

**Figure 8.** Annual difference in ozone production (red) / destruction (blue) flux between "New Het." and control runs, 1996–2009.

**Table 2.** Difference in heterogeneous reaction flux between "New Het." and control runs, 1996–2009.

| Reaction | Flux (mol/sec) | | | |
| --- | --- | --- | --- | --- |
| | Control | New Het. | Diff. | % Diff |
| $ClONO_2 + HCl$ | 468 | 747 | +279 | +59.6 |
| $ClONO_2 + H_2O$ | 1300 | 1040 | -266 | -20.4 |
| $HOCl + HCl$ | 1230 | 1190 | -46.7 | -3.8 |
| $N_2O_5 + H_2O$ | 8180 | 7880 | -296 | -3.6 |
| $N_2O_5 + HCl$ | 3.02 | 3.13 | +0.114 | +3.8 |
| $HOBr + HCl$ | - | 22.9 | - | - |
| $BrONO_2 + HCl$ | - | 981 | - | - |
| $BrONO_2 + H_2O$ | - | 560 | - | - |
| $HOBr + HBr$ | - | 13.1 | - | - |
| $HOCl + HBr$ | - | 0.167 | - | - |
| $ClONO_2 + HBr$ | - | 0.209 | - | - |
| $BrONO_2 + HBr$ | - | 0.509 | - | - |
| $N_2O_5 + HBr$ | - | 0.000158 | - | - |

Figure 9(a) shows the October SH polar mean vertical profile of the bromine species: $BrO$, $HBr$ and $BrONO_2$ in the "New Het." (solid curves) and control (dashed) runs. The effect on $BrONO_2$, suggested by the fluxes in Table 2, is shown to be quite substantial here. The lower peak in $BrONO_2$ at around 12 km is strongly reduced, with a co-located decrease in $BrO$ and increase in $HBr$ localized to the the same altitude range. Figure 9(b) shows the equivalent to Figure 9(a) for the chlorine

5    species: $ClO$, $Cl_2O_2$, $HCl$ and $ClONO_2$. The impact on the chlorine species varies over a larger altitude range. $ClONO_2$





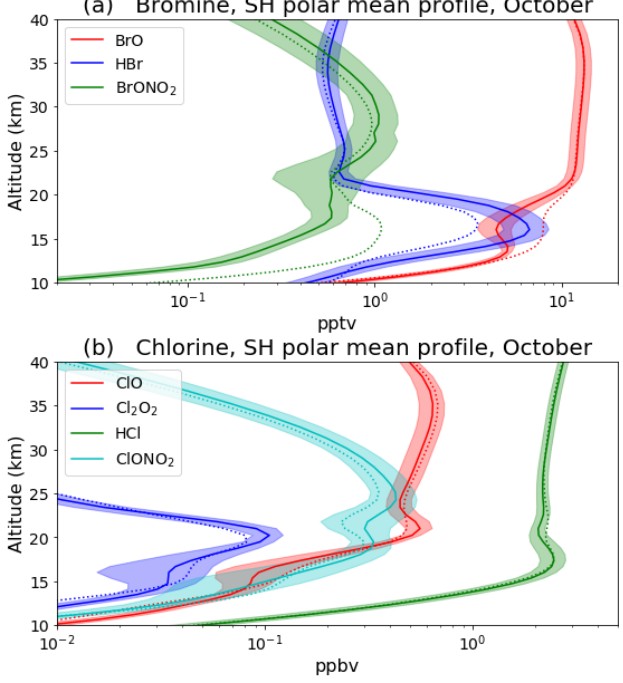

**Figure 9.** Vertical profile of (a) the species: BrO, HBr, and BrONO$_2$ and (b) ClO, Cl$_2$O$_2$, HCl, and ClONO$_2$ for October, averaged over 1996–2009 and south of 65°S. "New Het." is shown with solid curves and the control run with dashed curves. Shading indicates ±1 standard deviation around the "New Het." mean.

increases broadly over both peaks around 19 and 26 km, this is primarily due to the decrease uptake on NAT. Below about 15 km, where SA is more prominent, ClONO$_2$ is reduced due to the new loss pathway of reaction with HCl on SA.

## 4.3    Discussion

The changes to the heterogeneous chemistry result in increased SH springtime polar ozone depletion. This has the advantage
5   of more accurately simulating the timing and rate of decline of ozone in early spring. During October, the peak ozone depletion is slightly overestimated relative to the NIWA-BS data, however this is likely due to the model's cold bias (see Figure 6). The downside of this change occurs during summer where the model has a tendency to replenish the SH polar ozone too slowly. It is possible the increased ozone depletion itself exacerbates this problem as it has been shown that increased ozone depletion acts to delay the break-up of the polar vortex (Haigh and Roscoe, 2009; McLandress et al., 2010; Keeble et al., 2014). However, the
10   model runs presented here show no evidence of this effect, with the ozone replenishing at similar rates in either configuration. This summertime ozone problem could be both dynamical, for example: a lack of planetary wave activity which delays the vortex break-up, or chemical, for example: in the conversion of radicals back into reservoir species. This remains an important area for model development.



Another area of possible model development is the treatment of PSCs in the UKCA. In the case of NAT, particles are assumed to form when $HNO_3$ reaches saturation (following Hanson and Mauersberger (1988)). More recent work has found that NAT formation does not occur at the equilibrium temperature predicted by this assumption, but at temperatures around 3K lower (Wegner et al., 2013). This would suggest that a $HNO_3$ supersaturation of around 10 ought to be required for NAT

formation. It is possible that this may not be very important given that the role of NAT is somewhat de-emphasized in our new heterogeneous chemistry scheme, with the corresponding uptake coefficients on NAT lowered in four of the five original reactions and not considered for six of the additional eight reactions. Perhaps of more consequence is the treatment of sulfate aerosol. UKCA considers sulfate aerosol, however sulfate aerosol absorbs $HNO_3$ at lower temperatures, forming what is known as a supercooled ternary solution (STS) (Hamill et al., 1996). Data regarding reactions on STS are more limited, but it has been

shown to be important in some cases. For example, Zhang et al. (1995) found that the rate of $N_2O_5$ hydrolysis on STS to decrease as the amount of $HNO_3$ in solution is increased, while the rate of $ClONO_2$ hydrolysis is unaffected by the presence of $HNO_3$.

A hazard of modeling stratospheric chemistry is the limited experimental data on which parameterizations are based. Of the eight reactions added to the heterogeneous chemistry scheme, the most active of these are the reactions of $BrONO_2$

with HCl and $H_2O$ on sulphate aerosol. The experimental work on these reactions (Hanson and Ravishankara, 1995; Hanson et al., 1996; Hanson, 2003) focused mostly on the hydrolysis reaction. The uptake coefficient of 0.9 for $BrONO_2 + HCl$, recommend by JPL, is based upon just two experiments (Hanson et al., 1996). One tests 60 wt% $H_2SO_4$ with a small amount of HCl (approx. $10^{-3}$M), the other 48 wt% with 0.3M HCl, both at 229 K, finding uptake coefficients of 0.9±0.2 and 1.0±0.2, respectively. Hence, there may be substantial uncertainty associated with this value. Perhaps because of this uncertainty, this

reaction is not included in some chemistry-climate models; for example, the Whole Atmosphere Community Climate Model (WACCM) excludes this reaction while including the $BrONO_2 + H_2O$ reaction (Solomon et al., 2015). However, a coefficient as large as 0.9 is not necessarily surprising given that the parameterization of $BrONO_2 + H_2O$ (which is based on much more experimental data) gives an uptake coefficient of 0.8 for solutions with $H_2SO_4$ at less than 65%wt. As Table 2 shows $BrONO_2 + HCl$ to be relatively important, we choose to include it despite the associated uncertainty.

Compared to the $BrONO_2$ reactions with $H_2O$ and HCl the other additional reactions are shown to have relatively little impact. However, we include these in the UKCA heterogeneous chemistry framework for completeness, as it is possible that future work may reveal them to be more relevant. In particular, further investigation of these reactions on sulphate aerosols would be useful given that the uptake of $BrONO_2$ is known to be so large.

The sulfate aerosol used in this work is a climatology representing the "background" level, so the effect on ozone demon-

strated here could be perhaps thought of as a lower limit. Further work looking at the sensitivity of this heterogeneous chemistry scheme to the effect of elevated aerosol levels (e.g. from volcanic eruptions) would also be of interest.

Given the importance of bromine activation on sulfate aerosols, another interesting aspect is the sensitivity of ozone depletion to the abundance of bromine. Yang et al. (2014) examined this, also using the UM-UKCA model, finding a 5% decrease in springtime SH polar TCO (~10 DU) in response to an increase in stratospheric inorganic bromine resulting from a doubling

of the very short lived species (VSLS) source. The uptake coefficients used by Yang et al. (2014) differ from those used here;





this configuration shows a larger impact on ozone. It is therefore possible that the ozone depletion sensitivity demonstrated by Yang et al. (2014) is somewhat of an underestimation. Yang et al. (2014) also finds the ozone depletion sensitivity to bromine concentration to be dependent on the amount of chlorine present. Thus, a deeper study of the role of the chlorine/bromine cross reactions would be an interesting area for further study with this new configuration of heterogeneous chemistry.

## 5    Summary

Improvements to the UKCA model have been made in two areas: the photolysis and the heterogeneous chemistry schemes. The photolysis scheme was improved by the addition of solar variability which modifies the solar flux entering the photolysis rate calculations. The solar cycle is based on observations and is spectrally resolved. At short wavelengths, the amplitude of the solar cycle is of the order of 10%. Consequently, reactions (such as the photolysis of oxygen) that require this short-wavelength radiation vary proportionately. We show that the effect of the added solar cycle on total column ozone in the extra-tropical regions is of the order of 1%, in phase with the solar cycle. The effect is likely to be larger in polar regions although the model run in this study is not of sufficient length to provide reliable estimates given the increased variability of ozone in these regions.

The second improvement is to the heterogeneous chemistry scheme. This consisted of updating the uptake coefficients of the five pre-existing reactions as well as adding a further eight reactions. The impact on ozone from these changes were substantial, amounting to around 25 DU of additional springtime SH polar ozone depletion, with significant effect coming from both the updated original reactions and the new reactions. Comparison with the NIWA-BS ozone dataset shows that the new heterogeneous chemistry scheme simulates springtime ozone better, especially when accounting for the model temperature bias, although the increase in springtime ozone depletion leads to an increase in the low-ozone bias during summer.

*Code and data availability.*   The Met Office Unified Model is available for use under licence. For information on how to apply for a licence; see http://www.metoffice.gov.uk/research/ modelling-systems/unified-model (last access: 30 Oct 2018). Both developments are available in the UM truck from vn11.2. NIWA-BS total column ozone data (vn3.4) were obtained from http://www.bodekerscientific.com/data/total-column-ozone (retrieved 12 Oct 2018). ERA-Interim data were obtained from http://apps.ecmwf.int/datasets/data/interim-full-mnth/levtype=pl/ (retrieved 16 May 2018). Data from the UM-UKCA model runs and the code used to produce the figures in this paper can be found at https://doi.org/10.5281/zenodo.1486305

*Author contributions.*   FD wrote the paper with input from all authors. The solar cycle code was originally written by OM, with modifications by FD. The heterogeneous chemistry code was written by JK and FD with assistance from all authors. Model runs and analysis were produced by FD with advice from OM and GZ



*Competing interests.* The authors declare that they have no conflict of interest.

*Acknowledgements.* This work has been funded by the New Zealand Government Ministry for Business, Innovation, and Employment (MBIE) through The Deep South National Science Challenge. This work has also been supported by NIWA as part of its Government-funded, core research. Also, funding was received from European Community's Seventh Framework Programme (FP7/2007-2013) under

5    grant agreement no. 603557 (StratoClim). The authors wish to acknowledge the contribution of NeSI high-performance computing facilities to the results of this research. New Zealand's national facilities are provided by the New Zealand eScience Infrastructure (NeSI) and funded jointly by NeSI's collaborator institutions and through MBIE's Research Infrastructure programme (https://www.nesi.org.nz). We would also like to thank Bodeker Scientific, funded by the New Zealand Deep South National Science Challenge, for providing the combined NIWA-BS total column ozone database



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
