# Peer review of "Improvements to stratospheric chemistry scheme in the UM-UKCA (v10.7) model: solar cycle and heterogeneous reactions"

_Geoscientific Model Development, 2018_

## Referee Comment (RC1) · Anonymous Referee #1 · 20 Dec 2018

This paper presents improvements made to the stratospheric chemistry part of the UM-UKCA chemistry-climate model. First, the photolysis scheme is improved by accounting for the 11-year solar cycle variability. Second, the heterogeneous chemistry scheme is improved by including or updating reactive uptake coefficients on sulphuric acid aerosols and polar stratospheric clouds. The effects on total column ozone of these model improvements are assessed with a focus on the Antarctic. Some of the model ozone biases are found to be reduced by these improvements. Overall, the results are interesting and helpful to stratospheric ozone modellers. The paper is well written and clear. Therefore, I recommend publication. However, the authors might wish to consider and even take on board some of the questions, comments and sug-
gestions listed below.

* General comments

- It is difficult to understand why the photolysis and heterogeneous chemistry simulations have different set-ups. In the case of heterogeneous chemistry simulations: l9 p8: 'In order to examine only the direct effect of the modifications, feedbacks between the chemistry (ozone, methane, nitrous oxide and water vapour) and radiation have been switched off for these runs (i.e. these runs are identical in terms of their meteorological variables).' That is a very good set-up to check the effects of changes in the chemistry scheme without dynamical feedbacks. Then the authors do additional runs with fully coupled chemistry-dynamics (i.e. model-calculated fields of ozone, methane, nitrous oxide and water vapour used in the radiation scheme) to see the full effects on ozone. That is a very sensible approach. In contrats, in the photolysis simulation, only short (only 15 years for 11-years solar cycles) fully coupled simulations are run. As a result, the analysis and interpretation are difficult because of the dynamical variability (differences in temperatures and winds between the 2 runs). The authors are well aware of these limitations: l1, p5: 'We only plot the results for the 60S–60N region here because the variability in ozone is quite large in the polar regions, masking the impact of solar variability.' or l14, p5: 'Figure 2 suggests there may be some phase shift between the solar cycle and its effect on stratospheric ozone. Although, the model runs presented here are too short to verify'. The authors cannot conclude with this model set-up. I would suggest to run 2 additional simulations without the dynamical feedbacks (i. 'runs are identical in terms of their meteorological variables'). It means using up computing resources but the runs are short (15 years) and this would really help the analysis.

- The heterogeneous chemistry part contains a nice latitude and altitude-resolved analysis. I don't know why the photolysis part is so short with no latitude or altitude-resolved analysis. Again, I would suggest to follow not only the heterogeneous chemistry set-up but also the structure of the analysis. I am sure that readers would be interested by seeing the 11-year solar cycle signal in the global zonal mean ozone distribution.

[Figure]

- I would suggest to move subsections 3.1 and 4.1 into the very short section 2 which is devoted to model description. sections 3 and 4 should focus on presenting and discussing the results. This would facilitate the flow of the reading.

* Specific comments

l7, p2: Not correct. Solar radiation also varies on solar rotational timescales (about 27 days) with amplitude just slightly lower than the 11-year cycles. It also varies on longer time-scales.

l10, p2: not only early 2000s. Chipperfield et al, Nature, 2017 study (along with the supplementary information) pointed out the effects of the 11-year solar variability (not only about the early 2000s but also about the mid-2010s) on ozone trends.

l19, p2: constellations? configurations

l14, p3: "...using a combination of the FAST-JX scheme (Wild et al., 2000; Bian and Prather, 2002; Neu et al., 2007) and look-up tables." In terms of computing costs, it does not seem to be very efficient to combine a look-up table and an on-line photolysis model. Why not use either a look up table or Fast-J?

Figure 2, p5: solar variability does not simply impact the ozone production (molecular oxygen photolysis), it also impact ozone destruction via the change in Ox partitioning (notably the photolysis of O3 to atomic oxygen which is a key reactant in the ozone-destroying catalytic cycles. I would suggest to show also the O3-to-O photolysis.

Figure 2 caption: 'the flux of the oxygen photolysis'. Do you mean the solar flux below 240 nm, the solar flux below 240 nm weighted by the O2 cross sections, or O2 photolysis rate ?

l18-20, p6: 'It may be possible to extend our treatment of solar variability to the UM radiation scheme although it is unlikely to have a large effect given the smaller amplitude of the solar cycle at the relevant wavelengths.' for the temperature, it is not at all small Swartz et al., ACP, 2012. I know that the direct heating continuation to ozone changes

is much smaller but not negligible. It is better to provide a reference for this statement.

l14, p8: it is expected. remove 'as'.

ll9, p9: no need to be so cautious. Figure 6 (nice way to compare CCM with T biases and observations) proves it. It clearly demonstrates that, at least over the 210-215 K interval the new model performs much better in terms of Antarctic ozone October climatology. An hypothetical correction of T biases would be equivalent to running a nudged version of the model and surely getting better results with the new version.

l2, p11 and so on: I suggest to add chemical when talking about flux in order to avoid confusion with other fluxes (e.g. dynamical).

Figure 8 caption: is it global mean?

l4-5, p13: unclear to me. The bromine heterogeneous reactions convert BrONO2, HBr and HOBr into bromine radicals (Br and hence BrO). Why does BrO decrease and HBr increases?

l9-10, p13: Figure 5 shows the opposite. I think that the most likely explanation is that less O3 means less heating and hence longer lasting vortex. It is a classic example of an ozone bias resulting in a temperature bias.

l11-13, p13: I suppose that this statement is not just a guess. A reference for it?
* * *

---

## Referee Comment (RC2) · Anonymous Referee #2 · 24 Dec 2018

An update of the UM-UKCA stratopsheric chemistry scheme is presented in this paper. The heterogeneous reactions have been extended to also include bromine reactions on sulphuric acid aerosol and polar stratopsheric clouds. Furthermore the photolysis rate calculations include now the 11-year solar cycle variability. The latter impacts total column ozone only in the order of 1-2%.Whereas the additional heterogeneous reactions improve total column ozone in polar winter and spring, especially over Antarctica. Both updates take the model to a standard other chemistry-climate models covering the middle atmosphere have achieved more than a decade ago. Nevertheless, I recommend publication. The paper is thoroughly written and the impact of the changes are generally well documented and helpful to modellers in this field. Furthermore it

provides a documentation on the development of the UM-UKCA model.

* General comments

- I support the criticism of Referee #1 concerning the statistical significance of the solar cycle simulation. A closer look at figure 2 reveals, that up to the year 2000 ozone anomalies closely follow the solar coefficient. This signal is lost after 2000, where it could be interpreted as noise around the zero line. The author's provide no statistical evidence (and I think they can not) to convince the reader, that the model is able to show the impact of the solar cycle on total column ozone (TCO). What to do? Either extend the simulation, as suggested by referee #1, and hope that the signal on TCO can then be certainly identified, or use a different indicator. TCO might not be the best indicator for the impact of the solar cycle on stratospheric ozone. The signal is rather small. For instance a time series which does not only look at TCO, but considers the vertical distribution of stratospheric ozone could save the day. Also to discriminate between hemispheres or tropics and mid latitudes might help. There could be compensating effects (chemical or dynamical ones), which lead to the noisy signal in TCO. All in all, I strongly suggest a more sound statistical analysis of the impact of the solar cycle on stratospheric ozone.

* Specific comments, additional comments on topics, which have not been covered by Referee #1:

- l20-30, p2: A GMD paper does not need to provide a complete extensive overview on the field. But at least the fact that some chemistry-climate models have included the heterogeneous stratospheric bromine reactions for a long time should be mentioned, e.g. see supplement of Jöckel et. al. ACP, 2006 and others.

———————————————

---

## Author Response (AR1)

Reviewer #1

* General comments

- It is difficult to understand why the photolysis and heterogeneous chemistry simulations have different set-ups. In the case of heterogeneous chemistry simulations: l9 p8: 'In order to examine only the direct effect of the modifications, feedbacks between the chemistry (ozone, methane, nitrous oxide and water vapour) and radiation have been switched off for these runs (i.e. these runs are identical in terms of their meteorological variables).' That is a very good set-up to check the effects of changes in the chemistry scheme without dynamical feedbacks. Then the authors do additional runs with fully coupled chemistry-dynamics (i.e. model-calculated fields of ozone, methane, nitrous oxide and water vapour used in the radiation scheme) to see the full effects on ozone. That is a very sensible approach. In contrats, in the photolysis simulation, only short (only 15 years for 11-years solar cycles) fully coupled simulations are run. As a result, the analysis and interpretation are difficult because of the dynamical variability (differences in temperatures and winds between the 2 runs). The authors are well aware of these limitations: l1, p5: 'We only plot the results for the 60S–60N region here because the variability in ozone is quite large in the polar regions, masking the impact of solar variability.' or l14, p5: 'Figure 2 suggests there may be some phase shift between the solar cycle and its effect on stratospheric ozone. Although, the model runs presented here are too short to verify'. The authors cannot conclude with this model set-up. I would suggest to run 2 additional simulations without the dynamical feedbacks (i. 'runs are identical in terms of their meteorological variables'). It means using up computing resources but the runs are short (15 years) and this would really help the analysis.

*We now use two new 20-year runs with chemistry-radiation feedbacks turned off. Figure 2 now shows a global average TCO rather than 60N-60S average as high latitude variability is no longer a factor. These runs show a very strong correlation between the solar cycle and ozone. The paragraph on l9-15 p6 modified to reflect that the new runs no longer show the phase shift between the solar cycle and ozone.*

- The heterogeneous chemistry part contains a nice latitude and altitude-resolved analysis. I don't know why the photolysis part is so short with no latitude or altitude-resolved analysis. Again, I would suggest to follow not only the heterogeneous chemistry set-up but also the structure of the analysis. I am sure that readers would be interested by seeing the 11-year solar cycle signal in the global zonal mean ozone distribution.

*Now that I am using the no-feedback runs there is very little latitudinal variation. I have added a new Figure 3 which shows the variation with altitude. A paragraph (p5 ln 5-15) has been added to describe this figure.*

- I would suggest to move subsections 3.1 and 4.1 into the very short section 2 which is devoted to model description. sections 3 and 4 should focus on presenting and discussing the results. This would facilitate the flow of the reading.

*To me it makes more sense to have these sections separate. For example, it is more convenient having Table 1 and a description of the model runs immediately before the results as it serves as somewhat of an introduction to the section and saves the reader going all the way back to section 2 for a reminder of what the model changes were.*

* Specific comments

l7, p2: Not correct. Solar radiation also varies on solar rotational timescales (about 27 days) with amplitude just slightly lower than the 11-year cycles. It also varies on longer time-scales.

*L7, p2 Changed to: "One of the main modes of variation in solar radiation is the 11-year cycle…"*

l10, p2: not only early 2000s. Chipperfield et al, Nature, 2017 study (along with the supplementary information) pointed out the effects of the 11-year solar variability (not only about the early 2000s but also about the mid-2010s) on ozone trends.

*L12, p2 Added: "More recent increases in ozone have been shown to be above those expected by the next solar maximum and are therefore indicative of the effect of declining stratospheric chlorine [Keeble et al. 2018, Chipperfield et al. 2017]."*

l19, p2: constellations? configurations

*l20, p2: Changed to configurations*

l14, p3: ". . .using a combination of the FAST-JX scheme (Wild et al., 2000; Bian and Prather, 2002; Neu et al., 2007) and look-up tables." In terms of computing costs, it does not seem to be very efficient to combine a look-up table and an on-line photolysis model. Why not use either a look up table or Fast-J?

*The look-up table is necessary in the upper levels of the model where the shorter wavelengths are important. The way the model is set-up FAST-JX runs at all levels, so the option is either to add on the contribution from FAST-JX at the upper levels or not i.e. there is no computational benefit from only using the look-up table at these levels.*

Figure 2, p5: solar variability does not simply impact the ozone production (molecular oxygen photolysis), it also impact ozone destruction via the change in Ox partitioning (notably the photolysis of O3 to atomic oxygen which is a key reactant in the ozone destroying catalytic cycles. I would suggest to show also the O3-to-O photolysis.

*Changes in O3 loss are now shown in Figure 3. Pg 5 ln 5-15 have been added which describe the figure and note the impact of the solar cycle on the ozone photolysis rate.*

Figure 2 caption: 'the flux of the oxygen photolysis'. Do you mean the solar flux below 240 nm, the solar flux below 240 nm weighted by the O2 cross sections, or O2 photolysis rate ?

*Changed to "chemical flux"*

l18-20, p6: 'It may be possible to extend our treatment of solar variability to the UM radiation scheme although it is unlikely to have a large effect given the smaller amplitude of the solar cycle at the relevant wavelengths.' for the temperature, it is not at all small Swartz et al., ACP, 2012. I know that the direct heating continuation to ozone changes is much smaller but not negligible. It is better to provide a reference for this statement.

*I was basing this on Bednarz et al. 2018 which is mentioned in the following sentence. Changed that sentence to "Specifically, Bednarz et al. [2018] find that the difference between solar maximum and minimum in the shortest UM wavelength band (covering 200-320 nm) is only 0.56%, …" to make the connection clearer.*

l14, p8: it is expected. remove 'as'.

*L5, p9: Removed 'as'*

ll9, p9: no need to be so cautious. Figure 6 (nice way to compare CCM with T biases and observations) proves it. It clearly demonstrates that, at least over the 210-215 K interval the new model performs

much better in terms of Antarctic ozone October climatology. An hypothetical correction of T biases would be equivalent to running a nudged version of the model and surely getting better results with the new version.

*L6, p10: Removed "it is likely that a hypothetical"*

l2, p11 and so on: I suggest to add chemical when talking about flux in order to avoid confusion with other fluxes (e.g. dynamical).

*L10, p10: Changed to "chemical flux"*

Figure 8 caption: is it global mean?

*No. Caption changed to: "Annual difference in ozone production (red) / destruction (blue) flux between ``New Het.'' and control runs, in the lower stratosphere (12--24 km) south of 65S 1996--2009."*

l4-5, p13: unclear to me. The bromine heterogeneous reactions convert BrONO2, HBr and HOBr into bromine radicals (Br and hence BrO). Why does BrO decrease and HBr increases?

*I suspect this may be due to interaction between the Cl and Br chemistry. As BrONO2 loss is occurs mostly on sulphate aerosol in is quite widespread. However, the changes to HBr and BrO are localized to the SH high latitudes. I did not output the reaction fluxes necessary to fully investigate this (and a full investigate would likely be outside the scope of this paper).*

l9-10, p13: Figure 5 shows the opposite. I think that the most likely explanation is that less O3 means less heating and hence longer lasting vortex. It is a classic example of an ozone bias resulting in a temperature bias.

*L3, p14: Removed the sentence "However, the model runs presented here show no evidence of this effect, with the ozone replenishing at similar rates in either configuration."*

l11-13, p13: I suppose that this statement is not just a guess. A reference for it?

*These are just a couple of possibilities that I am suggesting.*

Reviewer #2

* General comments

 - I support the criticism of Referee #1 concerning the statistical significance of the solar cycle simulation. A closer look at figure 2 reveals, that up to the year 2000 ozone anomalies closely follow the solar coefficient. This signal is lost after 2000, where it could be interpreted as noise around the zero line. The author's provide no statistical evidence (and I think they can not) to convince the reader, that the model is able to show the impact of the solar cycle on total column ozone (TCO). What to do? Either extend the simulation, as suggested by referee #1, and hope that the signal on TCO can then be certainly identified, or use a different indicator. TCO might not be the best indicator for the impact of the solar cycle on stratospheric ozone. The signal is rather small. For instance a time series which does not only look at TCO, but considers the vertical distribution of stratospheric ozone could save the day. Also to discriminate between hemispheres or tropics and mid latitudes might help. There could be compensating effects (chemical or dynamical ones), which lead to the noisy signal in TCO. All in all, I strongly suggest a more sound statistical analysis of the impact of the solar cycle on stratospheric ozone.

*See response to Reviewer 1*

\* Specific comments, additional comments on topics, which have not been covered by Referee #1:

- l20-30, p2: A GMD paper does not need to provide a complete extensive overview on the field. But at least the fact that some chemistry-climate models have included the heterogeneous stratospheric bromine reactions for a long time should be mentioned, e.g. see supplement of Jöckel et. al. ACP, 2006 and others.

*L33, p2 Added: "Various heterogeneous bromine reactions have been included in other chemistry-climate models [e.g. Jockel et al. 2006, Wegner et al. 2013]."*

The following line numbers  refer to the position in the marked-up document.

P2 L7: "One of the main…"

P2 L12: "More recent increases in ozone have been shown to be above those expected by the next solar maximum and are therefore indicative of the effect of declining stratospheric chlorine (Keeble et al.,2018; Chipperfield et al., 2017)"

P2 L21: "…configurations…"

P2 L33: "Various heterogeneous bromine reactions have been included in other chemistry-climate models (e.g. Jöckel et al., 2006; Wegner et al., 2013)."

P4 L6: "20-year"

P4 L7: "In order to examine only the direct effect of the modification, feedbacks between the chemistry(ozone, methane, nitrous oxide and water vapour) and radiation have been switched off for these runs (i.e. these runs are identical in terms of their meteorological variables)."

P5 L7: "In addition to ozone production via oxygen photolysis, variations in solar flux also affect ozone loss pathways. These include the photolysis of ozone, a component of the Chapman cycle, and various catalytic loss cycles. We illustrate the effect of the solar cycle in Figure 3(a) which shows the difference in mean chemical fluxes of the ozone production (blue)and various loss pathways as functions of altitude for the year 1989 (i.e. near a solar maximum). The figure also shows the net effect from the changes in the production and loss (black curve). The photolysis of ozone extends to longer wavelengths than that of oxygen (Chapman, 1930) and hence is less influenced by the solar cycle than the photolysis of oxygen (at solar maximum the photolysis rate of ozone increases by around 1%). However, the various changes in the fluxes of the catalytic loss cycles offset the increase in production to a large extent. While the effect on ozone production peaks around 44 km, Figure 3(b) shows that the impact on the ozone mixing ratio peaks at a somewhat lower altitude, around 39 km. This is due, in large part, to theHO2loss cycle that increases strongly above 40 km. The effect on the ClO cycle and the loss branch of the Chapman cycle (i.e. the O3+O(3P) reaction) also peak slightly above 40 km.

P6: Figure 3 added

P6 L9: "shows the variation in ozone to be strongly in phase with the solar cycle. Previous work has found there to be phase shift between the solar cycle and its effect on ozone"

P6 L14: "Because the model runs used here have chemistry-radiation feedback switched off we would not expect to see this effect

P7 L10: "Specifically, …"

P7 L32: "Similar to the solar cycle runs, chemistry–radiation feedbacks are switched off for these runs in order to examine only the direct effect of the modifications"

P9 L5: removed "as", now reads: "This is expected…"

P10 L6: removed "it is likely that a hypothetical", now reads: "Based on the gradient of the ozone-temperature relationship shown here, a correction of the temperature bias…"

P10 L10: "… chemical …"

Figure 9 caption: "in the lower stratosphere (12–24 km) south of 65◦S"

[revised manuscript text omitted]